# The role of aquaporin-4 in optic nerve head astrocytes in experimental glaucoma

**Elizabeth Kimball**⊙*, **Julie Schaub, Sarah Quillen**⊙, **Casey Keuthan**⊙, **Mary Ellen Pease, Arina Korneva, Harry Quigley**

Glaucoma Center of Excellence, Wilmer Eye Institute, Johns Hopkins University, Baltimore, Maryland, United States of America

* fcone1@jhmi.edu

**Data Availability Statement:** All relevant data are within the paper and its Supporting Information files.

## Abstract

### Purpose

To study aquaporin channel expression in astrocytes of the mouse optic nerve (ON) and the response to IOP elevation in mice lacking aquaporin 4 (AQP4 null).

### Methods

C57BL/6 (B6) and AQP4 null mice were exposed to bead-induced IOP elevation for 3 days (3D-IOP), 1 and 6 weeks. Mouse ocular tissue sections were immunolabeled against aquaporins 1(AQP1), 4(AQP4), and 9(AQP9). Ocular tissue was imaged to identify normal AQP distribution, ON changes, and axon loss after IOP elevation. Ultrastructure examination, cell proliferation, gene expression, and transport block were also analyzed.

### Results

B6 mice had abundant AQP4 expression in Müller cells, astrocytes of retina and myelinated ON (MON), but minimal AQP4in prelaminar and unmyelinated ON (UON). MON of AQP4 nulls had smaller ON area, smaller axon diameter, higher axon density, and larger proportionate axon area than B6 (all p≤0.05). Bead-injection led to comparable 3D-IOP elevation (p = 0.42) and axonal transport blockade in both strains. In B6, AQP4 distribution was unchanged after 3D-IOP. At baseline, AQP1 and AQP9 were present in retina, but not in UON and this was unaffected after IOP elevation in both strains. In 3D-IOP mice, ON astrocytes and microglia proliferated, more in B6 than AQP4 null. After 6 week IOP elevation, axon loss occurred equally in the two mouse types (24.6%, AQP4 null vs. 23.3%, B6).

### Conclusion

Lack of AQP4 was neither protective nor detrimental to the effects of IOP elevation. The minimal presence of AQP4 in UON may be a vital aspect of the regionally specific phenotype of astrocytes in the mouse optic nerve head.

**Funding:** This work was supported in part by National Eye Institute in the form of PHS research grants awarded to HQ (EY 02120) and Wilmer Eye Institute (EY 0 1765) and by unrestricted support from Saranne and Livingston Kosberg and from William T. Forrester. The funders had no role in study design, data collection and analysis, decision to publish, or preparation of the manuscript.

**Competing interests:** The authors have declared that no competing interests exist.

# Introduction

Aquaporins (AQPs) are transmembrane water channels that are permeable to water and small solutes; such as glycerol and urea [1]. Three AQP types are known to be present in astrocytes [2]; AQP1, AQP4 [3, 4], and AQP9 (also referred to as aquaglyceroporin which is permeable to glycerol and other small uncharged solutes). In brain, astrocyte membranes facing pia, axons, and capillary connective tissue have AQP channels [5–7]. Brain astrocytes express AQP4 in both normal and pathological conditions, while AQP1 and 9 are expressed in pathological states [8, 9]. AQP channels are transmembrane tetramers of aquaporin subunits, depending for their formation on the presence of membrane α- and β-dystroglycan (DG) [1, 10, 11]. DGs are linked to extracellular glycoproteins, such as agrin, and connect intracellularly to the cytoskeleton [12, 13]. The AQP4 channel is selectively permeable to water and possibly to dissolved gases [1].

Lamina cribrosa astrocytes line the connective tissue beams of large mammal and human optic nerve heads (ONH), while in rodents the astrocytes of the lamina span the ONH, with minimal connective tissue support. Both rodent and primate ONH astrocytes are connected to the peripapillary sclera via their basement membrane (BM), reinforced by junctional complexes. Indirect clinical and epidemiological evidence implicates nutritional deficit as an important feature of glaucoma pathogenesis [14]. The responses of ONH astrocytes are important to both nutrition and fluid balance in glaucoma pathogenesis [15, 16]. In mouse glaucoma, axonal injury is first detected in the unmyelinated optic nerve (UON) region at and immediately posterior to UON junction with choroid and sclera [17], just as glaucomatous axon injury begins at the corresponding unmyelinated lamina cribrosa zone in monkey and human eyes [18, 19].

In all mammalian ONHs [5–7], astrocytes are separated from capillary lumens by the capillary cytoplasm (with tight junctions), the capillary BM, an intermediate connective tissue, and the astrocytic BM. Astrocytes completely segregate axons from the vascular and peripapillary scleral spaces. Nutrients and water from the capillaries and astrocytes reach axons by flowing through astrocytic transmembrane channels, including monocarboxylate transporters [20–22], glucose transporters [21], and AQP channels.

Interestingly, aquaporins, including AQP4, are not present in the UON of the ONH in normal mouse [23, 24], rat [4], and dog [25], though they are found in the retina, superficial ONH, and myelinated nerve (MON) and retinal pigment epithelium [26]. This is critically important in the pathogenesis of glaucomatous optic neuropathy, since IOP-generated stress leads to axonal transport obstruction at the lamina cribrosa of primates and similarly in the UON of rodents. Previous AQP investigations refer to its presence in "optic nerve" (ON), most often without specifically detailing from which region(s) of the eye or optic nerve the tissue was obtained [27, 28]. Genetic deletion of AQP4 leads to increased brain extracellular volume [29] and AQP4 was downregulated in the ONH region in a chronic rat glaucoma model [30]. Glaucoma [31] or aging [32] may involve abnormal AQP function. Immunolabeling of human glaucoma eyes found reduced AQP9 in retinal ganglion cells, but presence or change in AQP4 in the ONH was not described [33]. Fluid movement [34] through the optic nerve and cerebrospinal fluid entry into the optic nerve is reportedly impaired in experimental glaucoma [35]—possibly due to alterations in AQP channel function. However, past studies did not specifically discuss at what level the normal and altered fluid movements occurred. Nor has the potential effect of AQP knockout on the extent of glaucoma damage been tested.

In this report, we studied the local distribution of AQP1, AQP4 and AQP9s in retina and optic nerve of mouse eyes. We tested the hypothesis that AQPs were critical to glaucoma damage by study of genetic deletion of AQP4 in short and medium-term glaucoma models in mice.

## Methods

### Animals

We included 173 animals in this study. All protocols were in accordance with the guidelines of the ARVO Statement for the Use of Animals in Ophthalmic and Vision Research, and approved and monitored by the Johns Hopkins University School of Medicine Animal Care and Use Committee. We studied 3 strains of mice, ranging from 2–9 months of age at the start of experiments: C57BL/6 (B6, Cat # 0664, Jackson Laboratories, Bar Harbor, ME, USA), Aquaporin 4 constitutive knockout (AQP4 null, B6 background, courtesy of Drs. Peter Agre and Michael Levy, Johns Hopkins School of Medicine, Baltimore, MD, USA- equal distribution of females and males) and wild-type non-fluorescentFVB/N-Tg(GFAP-GFP)14Mes mice, WT GFAP-GFP, Jackson Laboratories #003257, Bar Harbor, ME, USA,). To confirm the knockout of the AQP4 gene, mouse genotypes were determined by Transnetyx (Cordova, TN) using validated primers in wild type B6 (forward) 5′–ACTGGTTTCTGTCCAAAACTACACA, (reverse) 5′–ACGAATATGTATTTAGCTGGGCGTTA (intron between exons 1 and 2, should not amplify in nulls) and in AQP4 nulls (forward) 5′– TCTGTCCAAAACTACACATGAAAAGTGT, (reverse) 5′– GCTGTGGTGACAATGGCATAAAC (amplicon stretches across the entire deleted region). In these 3 mouse strains, we measured IOP, counted ON axon number and size, labeled AQPs and related molecules, assessed axonal transport blockade of amyloid precursor protein (APP), evaluated ON ultrastructure by transmission electron microscopy (TEM), and assessed loss of retinal ganglion cell axons after chronic IOP elevation induced by microbead injection (Table 1). Tissue was analyzed at the following time points after IOP elevation: 3 days (3D-IOP), 1 week (1W-IOP), or 6 Weeks (6W-GL).

Previously published [36] IOP data on B6 mice from our laboratory (n = 73) were utilized to compare to new B6 mouse data and to AQP4 data in IOP, axial length, and ON axon counts.

### Anesthesia, IOP elevation, and IOP measurements

A total of 153 mice underwent IOP elevation by bead injection, using a previously published protocol [37]. IOP elevations lasting 3 days, 1 week, or 6 weeks were produced by unilateral anterior chamber microbead injections. For injections and euthanasia, mice were anesthetized with an intraperitoneal injection of ketamine (50 mg/kg, Fort Dodge Animal Health, Fort Dodge, IA), xylazine (10 mg/kg, VedCo Inc., Saint Joseph, MO), and acepromazine (2 mg/kg, Phoenix Pharmaceuticals, Burlingame, CA) and received topical ocular anesthesia (0.5% proparacaine hydrochloride eye drops, Akorn Inc. Buffalo Grove, IL, USA). For IOP measurements independent of additional procedures, animals were anesthetized using a Rodent

**Table 1. Mouse groups.**

| Strain | Acronym | N | Age (months) | Sex (F:M) | Study |
|---|---|---|---|---|---|
| C57BL/6 | B6 | 76 | 4 | All F | IOPs and Optic Nerve Counts (Naïve, 6W-GL) |
| | | 18 | 2–3 | All F | LSM (3D-IOP, 1W-IOP)- Immunostaining |
| | | 10 | 2–3 | All F | TEM (naïve, 1W-IOP) |
| Aquaporin 4 Null | AQP4 null | 33 | 4 | 15:18 | IOPs and Optic Nerve Counts (Naïve, 6W-GL) |
| | | 16 | 2–3 | 8:8 | LSM (3D-IOP, 1W-IOP)- Immunostaining |
| | | 10 | 2–3 | 5:5 | TEM (naïve, 1W-IOP) |
| FVB/N-Tg(GFAP-GFP)14Mes | WT GFAP-GFP | 10 | 5–9 | 8:2 | Gene Expression |

IOP = intraocular pressure, LSM = laser scanning microscopy, TEM = transmission electron microscopy

3D-IOP = 3 day IOP elevation, 1W-IOP = 1 week IOP elevation, 6W-GL = 6 week glaucoma model.

Circuit Controller (VetEquip, Inc., Pleasanton, CA, USA) delivering 2.5% of isoflurane in oxygen, 500cc/minute.

One anterior chamber was injected with Polybead Microspheres (Polysciences, Inc., Warrington, PA, USA), suspended in sterile PBS (phosphate buffer saline solution), consisting of 2 µL of 6 µm diameter beads, then 2 µL of 1 µm diameter beads, followed by 1 µL of viscoelastic compound (10 mg/ml sodium hyaluronate, Healon; Advanced Medical Optics Inc., Santa Ana, CA). Injections were made with a 50 µm tip diameter glass cannula, connected to a Hamilton syringe (Hamilton, Inc., Reno, NV). The glass cannula was kept in place for 2 minutes to prevent the egress of beads after withdrawal. The contralateral eye was used as control. No animal was excluded from our study once the bead injection was completed.

IOP measurements were made using the TonoLab tonometer (TioLat, Inc., Helsinki, Finland), recording the mean of 6 readings with optimal quality grading. For microbead-injected animals, IOP was measured at the following time points: prior to injection, at 1 day, 3 days, 1 week, 2 weeks, and 6 weeks after injection (depending upon length of survival). The positive integral IOP value was calculated as the difference between the bead-injected eye and its fellow eye, integrating the sum of the area under a curve that expressed the difference in IOP between the eyes in units of mm Hg—days, but including only those times when IOP was higher in the bead- treated eye.

## Sacrifice and tissue preservation

For TEM studies, 20 mice (10 B6 and 10 AQP4 nulls) were euthanized by exsanguination under general intraperitoneal anesthesia, followed by intracardiac perfusion for 3 minutes with 4% paraformaldehyde in 0.1 M sodium phosphate buffer ($Na_3PO_4$, pH = 7.2), 1 minute of 0.1M cacodylate buffer, and 7 minutes of 2% paraformaldehyde/2.5% glutaraldehyde in cacodylate buffer. A cautery mark on the superior cornea provided subsequent orientation. Eyes were enucleated and stored in fixative overnight before dissecting and removing the anterior chamber. In some eyes, the ON was removed 1.5 mm behind the globe and separately processed, while in other eyes posterior cups were processed with their proximal ON attached for epoxy embedding. Tissue was post fixed in 1% osmium tetroxide ($OsO_4$), dehydrated in ascending alcohol concentrations, and stained in 1% uranyl acetate in 100% ethanol for 1 hour. Tissues were embedded in epoxy resin mixture at 60˚C for 48 hours.

Thirty-four mice processed for cryopreservation (18 B6 and 16 AQP4 null) were euthanized by exsanguination under general intraperitoneal anesthesia, followed by intracardiac perfusion for 10 minutes using 4% paraformaldehyde in 0.1 M sodium phosphate buffer ($Na_3PO_4$, pH = 7.2). The superior cornea was marked with cautery and eyes were enucleated and postfixed in 4% paraformaldehyde for 1 hour, then placed into phosphate buffer.

The globes were cleaned of muscle and fat, and the anterior globe and lens were removed. ONs were cut 1.5mm behind the globe and embedded in epoxy for ON axon analysis. The posterior globe with initial nerve segment was placed in ascending concentrations of sucrose in 0.1M $PO_4$ buffer followed by embedding in 2 parts 20% sucrose buffer to 1 part Optimal Cutting Temperature compound (OCT; Sakura Finetek USA. Inc., Torrance, CA). Samples were frozen with dry ice cooled 2-methylbutane and stored at -80˚C until sectioning in longitudinal orientation.

## Immunolabeling

Posterior poles of B6 and AQP4 null mice were cryosectioned in longitudinal orientation between 8 to 16 µm thick, and collected onto slides (Superfrost Plus; Fisher Scientific; Pittsburgh, PA) for storage at –80˚C before immunolabeling. Antibodies are listed in Table 2.

**Table 2. Primary and secondary antibodies.**

| Primary Antibody | Company | Catalog | Species | Dilution | Mono or Polyclonal |
|---|---|---|---|---|---|
| α-Dystroglycan | Sigma | 05–593 | Mouse | 1:200 | Monoclonal |
| Aquaporin 1 | Abcam | ab15080 | Rabbit | 1:500 | Polyclonal |
| Aquaporin 4 | Alomone Labs | 249–323 | Rabbit | 1:500 | Polyclonal |
| Aquaporin 9 | Millipore | AB3091 | Chicken | 1:500 | Polyclonal |
| Amyloid Precursor Protein | Invitrogen | 51–2700 | Rabbit | 1:125 | Polyclonal |
| DAPI | Roche | D9542 | Stain | 1:1,000 | |
| Glial Fibrillary Acidic Protein | Invitrogen | 13–0300 | Rat | 1:1,000 | Monoclonal |
| Ki67 | Abcam | ab15580 | Rabbit | 1:200 | Polyclonal |
| Iba1 | Abcam | Ab5076 | Goat | 1:250 | Polyclonal |
| **Secondary Antibody** | **Company** | **Catalog #** | **Species** | **Dilution** | |
| Laser 488 | Invitrogen | A11008 | Goat anti-rabbit | 1:200 & 1:500 | |
| Laser 488 | Invitrogen | A11006 | Goat anti-rat | 1:500 | |
| Laser 488 | Invitrogen | A21206 | Donkey anti- rabbit | 1:500 | |
| Laser 555 | Invitrogen | A32816 | Donkey anti-goat | 1:500 | |
| Laser 568 | Invitrogen | A11004 | Goat anti-mouse | 1:500 | |
| Laser 647 | Invitrogen | A32933 | Goat anti-chicken | 1:500 | |
| Laser 647 | Invitrogen | A21244 | Goat anti-rabbit | 1:500 | |

All antibodies have been validated and references can be found on individual company website.

Sections were blocked with normal goat serum (NGS, 2–10%)/ 0.1% BSA or normal donkey serum (NDS, 2%)/ 0.1% BSA in PBS for 30 minutes, rinsed, and co-incubated with primary antibody overnight at 4˚C. Tissues were washed and the secondary antibodies applied (dilutions listed below) with 4'-6-diamidino-2-phenylindole (DAPI: Cat # 10-236-276-001, Roche Diagnostics, Indianapolis, IN) at 1:1,000 for 1 hour, followed by washing and mounting in DAKO mounting media (cat# 23023, Agilent Technologies, Santa Clara CA). Images were collected on the Zeiss 710 or 510 laser scanning microscope (LSM, Carl Zeiss Microscopy, LLC, Thornwood, NY), using the Plan-Apochromat 20x/0.8 M27 objective, or 40x Plan Apochromat with separate tracks for each laser (405, 488, 555, 568 and/or 647 nm).

Sections labeled with anti-Ki67 measured cell proliferation. The control and IOP elevation, longitudinal ON sections were obtained. Ki67 positive nuclei within each of the 4 regions denoted above (Fig 1A) were counted manually to calculate density (positive cells/mm$^2$). Double labeling was performed on some sections to determine whether the Ki67 positive nuclei associated with astrocytes (expression of glial fibrillary acidic protein, GFAP, ten B6 control samples and and ten B6 3D-IOP samples) or with microglia (expression of ionized calcium binding adaptor molecule 1, Iba1, nine AQP4 null control samples and nine AQP4 null 3D-IOP samples). Actin was labeled by the probe Phalloidin 568 (catalog # A12380, Invitrogen/Thermo Fisher, Waltham, MA).

### Amyloid precursor protein imaging and analysis

Cryopreserved optic nerve sections were labeled with an antibody to amyloid precursor protein (APP) to quantify transport blockade, an early assessment of axonal injury in 3D-IOP eyes. LSM 710 images of APP labeled sections were processed in the FIJI package [38]. Using the polygon selection tool, the following regions of interest were outlined: retina, prelaminar ONH, UON and MON. The width of the PL region was set to be 25 μm wider than the end of BMO on each side. The retinal region was separated from the PL by a 25 μm space on each

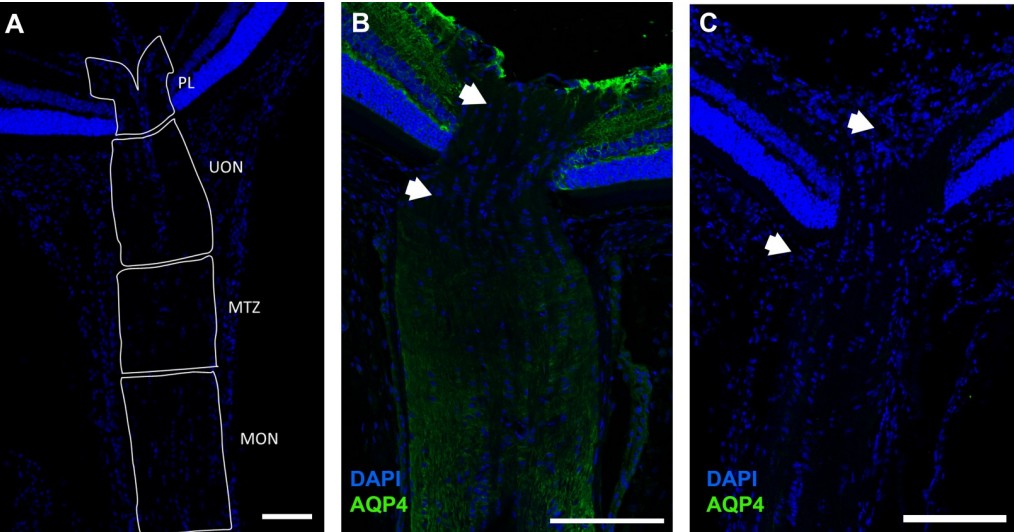

**Fig 1.** Longitudinal sections of C57BL/6 mouse (A,B), and Aquaporin 4 null (AQP4 null) mouse (C). Cryopreserved mouse optic nerve head tissues were labeled with anti-AQP4 (green) and DAPI (blue), then divided into 4 regions (A): pre-lamina (PL, from vitreoretinal surface to a line joining the two endpoints of BMO), unmyelinated optic nerve (UON, from BMO to 200 μm posteriorly), myelin transition zone (MTZ, from 200 μm to 350 μm posteriorly), and the myelinated optic nerve (MON, from 350 μm to the end of the section). Minimal label for AQP4 is visible in PL and the anterior portion of the UON in B6 mouse (between white arrows). AQP4 label is prominent in the myelinated portion of optic nerve in each species, as well as in retina. No AQP4 seen in the AQP4 null tissue (C). Scale Bar: 100 μm (A), 50 μm (B,C,E,F), 200 μm (E).

side and only the nerve fiber layer was included in retinal regions. The left and right retinal segments on each image were summed. Otherwise, the same regional marking was used as in the Ki67 analysis: PL, UON, and MON, but not including the MTZ for APP measurement. The APP fluorescence at 488 nm was exported in grayscale format with intensity values from 0 (black) to 255 (brightest value) pixels. To measure the effect of elevated IOP on APP blockade, we calculated the distribution of intensity, as histograms of pixel brightness in PL, UON and MON. In MATLAB (The Mathworks Inc. Natick MA), the pixel intensity values for each outlined region in each sample were extracted and several metrics were calculated: mean intensity in each region, and for each region the mean intensity and fraction of brightest pixels that exceeded the 97.5th percentile (%ile) of the normal IOP eye histogram. The latter two parameters were found better to express the degree of APP localized accumulation of material during axonal transport block in a previous publication [37]. These two parameters were calculated from the formulas: 1) Fraction > 97.5th%ile = (# pixels > T)/(total # pixels), and 2) mean > 97.5th%ile = (sum of pixels whose intensity was > T) / total # pixels), where T = the 97.5th %ile intensity value of the pixel distribution of the corresponding region in the matching normotensive control group.

## Quantification of aquaporin 4 fluorescence

Cryopreserved optic nerve sections of B6 controls (N = 4), and AQP4 null controls (N = 2) were analyzed for AQP4 brightness by measurement of pixel intensity value (PIV) using FIJI software for anti-AQP4 labeled images. Each image was set to 8-bits, limiting the PIV range from 0 to 255. In each region and zone the background was measured in 2 ways: 1) over a portion of the slide with no tissue, and 2) over the posterior choroid which lacks aquaporins. First, assessment of the regions was carried out in B6 controls. The UON was divided lengthwise

into two 100 μm long regions, one closer to the eye (the anterior-UON, from the BMO to 100 μm), and one in the more posterior unmyelinated nerve segment (the posterior-UON, from 100 μm to 200 μm). Both regions were within the unmyelinated portion of the nerve, prior to the MTZ. For each region, measurements were made in 3 zones: 1) a zone of peripheral ON (outer area, S1A Fig); 2) a zone more centrally (inner area); and 3) the total area (S1B Fig). We calculated the mean and median PIV in the overall optic nerve and in the inner and outer zones, as well as in the choroid, the PL, the MTZ and the MON. Once the regions and the method were established, PIV values were measured at 7 specific regions between the B6 and AQP4 samples, directly to compare values in choroid, retina, PL, anterior-UON, posterior UON, MTZ and MON.

## Quantitative real-time PCR (qRT-PCR)

Ten naïve 5–9 month old WT GFAP-GFP mice were euthanized with general intraperitoneal anesthesia. Eyes were enucleated, rinsed in cold PBS and the nerve was cut flush to the globe. Three different tissues were studied: retina, UON and MON. The UON portions were ~200 μm in length. The next 200–300 μm portion of nerve containing the MTZ was discarded. The MON was the first myelinated zone posterior to the MTZ, 200–300 μm in length. The anterior chamber was excised and the retina was removed. The brain tissue was extracted, rinsed in cold PBS, and sliced along the coronal and sagittal planes, producing segments of brain cortex. The dissected tissue was immediately placed in QIAzol Lysis Reagent (Qiagen) on ice. Three samples were collected for nerve and retinal tissue, along with 8 segments of brain tissue.

All samples were homogenized and RNA was isolated using standard QIAzol purification according to the manufacturer's protocols. cDNA was synthesized from the purified RNA using the High Capacity cDNA Reverse Transcription kit (Applied Biosystems) following the manufacturer's protocol. qRT-PCR was performed on a CFX384 Touch Real-Time PCR Detection system (Bio-Rad) with SsoAdvanced Universal SYBER Green Supermix (Bio-Rad). cDNA was diluted prior to PCR to obtain a final concentration of approximately 500 pg cDNA per reaction, with all reactions performed in triplicate. Primers were designed to span exon-exon boundaries, and all primer sets were validated by standard curve and melt curve analysis prior to experimental use. Amplification efficiencies of the validated primer sets ranged from 88–104%. PCR conditions were as followed: 95˚C for 2 min, followed by 40 cycles of 95˚C for 30 s and 60˚C for 30 s. Relative gene expression was calculated in reference to the geometric mean of the sample's corresponding housekeeping values (*Rpl19*, *Gapdh*, and *Actb*). The target genes measured were: *Aqp4*, *Gfap*, and *Cd68* (microglia marker). The primer sequences for each housekeeping and target gene were as followed: *Rpl19* (forward) 5′– `TCACAGCCTGTA CCTGAA`, (reverse) 5′- `TCGTGCTTCCTTGGTCTTAG`; *Gapdh* (forward) 5′– `CCAATGTGT CCGTCGTGGATC`, (reverse) 5′– `GCTTCACCACCTTCTTGATGTC`; *Actb* (forward) 5′– `AC CTTCTACAATGAGCTGCG`, (reverse) 5′–`CTGGATGGCTACGTACATGG`; *Gfap* (forward) 5′– `CAGAGGAGTGGTATCGGTCTAA`, (reverse) 5′–`GATAGTCGTTAGCTTCGTGCTT`; *Aqp4* (forward) 5′– `CCCGCAGTTATCATGGGAAA`, (reverse) 5′–`CCACATCAGGACAGAAGACA TAC`; *Cd68* (forward) 5′– `CCCACCTGTCTCTCTCATTTC`, (reverse) 5′– `GTATTCCACCGC CATGTAGT`. An ordinary one-way ANOVA with Tukey post-hoc test for multiple comparisons was performed to obtain p-values on the averaged data. A p-value $< 0.05$ was considered statistically significant.

## Transmission Electron Microscopy (TEM)

One-micron thick epoxy sections were cut in the retina and ON tissue, either perpendicular to the optic axis (cross-sections) or parallel to it (longitudinal sections) and stained with 1%

toluidine blue. Sections were imaged using Axiocam and Axioskop Imaging Software (Axios Media Inc., Arlington VA) at 10x to 63x. Ultrathin sections (~68 nm) were placed on copper grids and stained with uranyl acetate and lead citrate before being examined with a Hitachi H7600 TEM (Hitachi High Technologies, Clarksburg, MD). TEM images of the PPS/astrocyte border were taken from 4,000 to 30,000 times magnification. For measurement of mean myelin sheath thickness, TEM images of the ON were taken of B6 mice (at 21 locations, 200 axons in n = 4 ON) and AQP4 null mice (at 16 locations, 148 axons, in n = 3 ON).

### Optic nerve axon counts

One-micron thick cross-sections of the 6W-GL ONs were stained with toluidine blue and low power digital images of the nerves were used to measure ON area. Images were taken at 100x using a Cool Snap camera and analyzed with Metamorph software. For each nerve, five 40 x 40 $\mu m^2$ fields were acquired, corresponding to a 9% sample of total nerve area. Masked observers edited non-axonal elements from each image, generating an axon density from the software. The average axon density/mm$^2$ was multiplied by the individual nerve area to estimate total axon number. The software also calculated mean axon diameter. Axon loss was calculated by comparing 6W-GL eyes to the mean axon number in pooled, fellow eye nerves of the appropriate mouse strain.

### Statistical analysis

Data were tabulated and compared between treatment groups as mean ± standard deviation or median values. Statistical testing was performed. We used paired or unpaired t tests for normally distributed data or Wilcoxon rank sum tests for data failing normality testing (GraphPad Prism Version 8 (GraphPad Software Inc., La Jolla, California, USA), with significance level of p $\leq$ 0.05. Simple linear regression models were used to look at the effect of strain on percent axon loss with individual IOP exposure as co-variate. Model statistical analyses were performed using SAS 9.4 (SAS Institute, Cary, NC).

## Results

### IOP increase in bead-injected mice

Prior to bead injection, mean IOP in AQP4 null mice was not different from wild-type B6 mice: 10.4 ± 3.9 vs. 10.5 ± 3.5 mmHg, respectively (p = 0.92, unpaired t-test; includes IOP data from previously published B6 mice [36]). IOP increased significantly after bead injection compared to IOP in contralateral control eyes in both B6 and AQP4 nulls from 1–14 days after injection (p $\leq$ 0.01 for all, unpaired t-test). Mean IOP 1 day after bead injection was somewhat higher in B6 mice (27.7 ± 8.5 mmHg) than AQP4 null (24.1 ± 8.1 mmHg, p = 0.03, unpaired t-test). The mean IOP paired differences (treated minus untreated eye) from 1 day to 2 weeks after injection were not significantly different between AQP null and B6 eyes (Fig 2). For example, mean IOP at 3 days in B6 was 21.9 ± 9.9 mm Hg vs. 22.4 ± 5.6 mmHg in AQP null (p = 0.42, unpaired t-test). After 6W-GL, IOP exposure was numerically, but not significantly higher in AQP nulls than in B6 (positive integral IOP = 207.4 ± 218.1 mmHg/days AQP null vs. 152.3 ± 148.3 mmHg/day B6, p = 0.09, unpaired t-test). Peak IOP difference (treated minus control) was slightly, but not significantly higher in AQP4 null mice than B6 (16.5 vs. 12.7 mm Hg, p = 0.22, t-test).

### Localization of aquaporin 4 and α-dystroglycan

LSM images of control B6 (Fig 1B) mice demonstrated abundant AQP4 expression in retinal Müller cells, retinal nerve fiber layer astrocytes, and MON astrocytes. Minimal AQP4 labeling

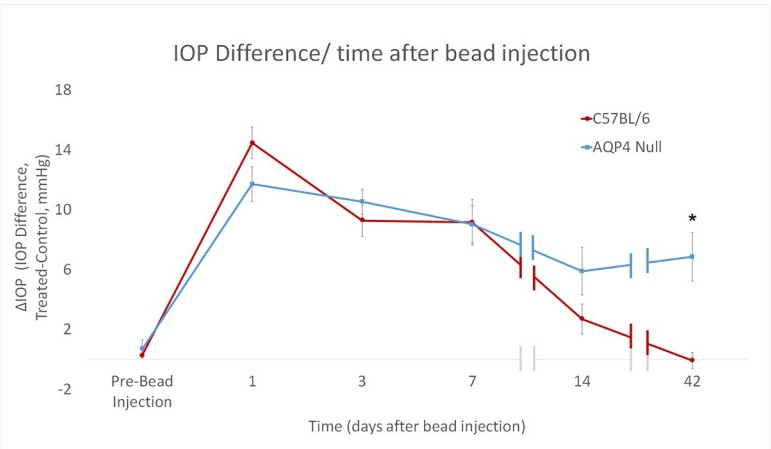

**Fig 2. The intraocular pressure (IOP) difference (ΔIOP) between bead-injected and untreated control eyes at 1, 3, 7, 14 and 42 days (hatch marks indicate discontinuities in time axis).** C57BL/6 (B6, red, N = 90) and aquaporin 4 null (AQP4 null, blue, N = 52). Standard error bars shown. Treated = microbead injected, control = contralateral untreated eye. *p ≤ 0.05.

was found in the PL and the anterior UON. Overall, AQP4 labeling was minimally present in the posterior UON region, and substantially greater in the MTZ and MON. There was no AQP4 labeling in any region of the AQP4 null eyes (Fig 1C).

AQP4-stained B6 mouse control pixel intensity values (PIV) of the outer area, inner area, and total area were compared (S1 Fig). The regional distribution of AQP4 was similar among the three areas. PIV from the total area, in each of the 7 regions, was compared between

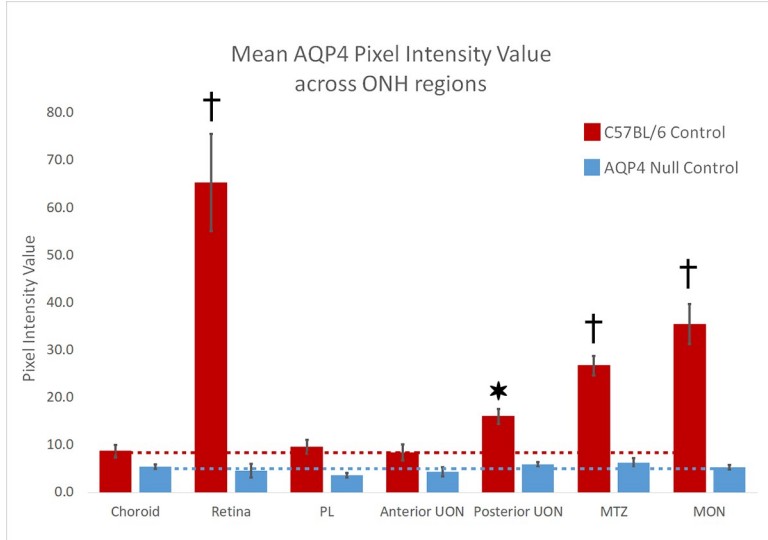

**Fig 3. Mean AQP4 Pixel Intensity Value (PIV) was calculated in C57BL/6 control (red bars, n = 2) and aquaporin 4 null (AQP4 null, blue bars, n = 4) samples using FIJI software.** Seven regions were analyzed, choroid (as our AQP4 negative control), retina, pre-lamina (PL, from vitreoretinal surface to a line joining the two endpoints of BMO), unmyelinated optic nerve was broken up into two sub regions, the anterior-UON (BMO to 100 μm), and posterior-UON (100 μm to 200 μm), myelin transition zone (MTZ, from 200 μm to 350 μm posteriorly), and the myelinated optic nerve (MON, from 350 μm to the end of the section). Standard error bars are plotted. Representative dotted lines identify the AQP4 background level in choroid for each mouse strain. *p ≤ 0.05, †p ≤ 0.001 for difference from control choroid.

control B6 and AQP4 nulls to quantify the presence of AQP4 throughout the retina and ON (Fig 3). In B6 mice, AQP4 PIV was highest in retina, 65.3 ± 15.0 (significantly higher than choroid PIV value of 8.7 ± 2.8, p ≤ 0.001, t-test, Fig 3) and lowest in PL and anterior UON regions (9.6 ± 3.1, and 8.4 ± 3.2; not significantly different from choroid, p = 0.67 and p = 0.90, unpaired t-test, respectively). Posterior-UON, MTZ, and MON PIV values were significantly higher than the choroid, PL, and anterior-UON regions (p ≤ 0.05, for all, unpaired t-test, Table 3). PL and anterior-UON regions were not significantly different from one another (p = 0.61, unpaired t-test). AQP4 null samples had no detectable AQP4 fluorescence above choroid background level (mean PIV ≤ 6, across all regions). In B6 control eyes, AQP1 was visible in the sclera, ON dura mater, in the nerve fiber layer and retinal ganglion cell (RGC) layer of the retina and in the MON (Fig 4A–4C). AQP1 and AQP9 were not present in UON in control B6. In control AQP4 null mice, AQP1 was found in the MON, sclera and ON dura mater, but not in the UON. AQP9 was detected only near blood vessels (Fig 4G–4I).

In B6, AQP4 label remained absent in UON after 3 day IOP elevation (3D-IOP, not shown), while AQP1 label was more prominent in the retinal ganglion cell layer, but unchanged in other regions (Fig 4D–4F). After 3D-IOP, AQP4 null eyes had increased AQP1 labeling at the RGC layer (as in B6 mice), but there was no detectable increase in label of AQP1 or AQP9 in the UON or in other regions of the ON (Fig 4J–4L).

Control mouse eyes had immunolabeling of αDG in the internal limiting membrane of retina, as well as in the PL, UON, MTZ and MON (Fig 5B). Specifically, αDG was present in the UON, which was devoid of AQP4 label in mouse eyes (Fig 5A and 5C).

## Regional gene expression difference in control mice

*Aqp4* mRNA was nearly undetectable in the UON (Fig 6A), in agreement with immunolabeling. Interestingly, *Aqp4* gene expression was significantly higher in the MON compared to all other tissues analyzed (UON, retina, and brain, p < 0.0001; Fig 6A). Differences in *Cd68* expression were very low and similar among the various tissues, including no significant difference between UON and MON regions (Fig 6B). *Gfap* mRNA expression (found in both astrocytes and retinal Müller cells) appeared higher in both UON and MON, with significantly greater expression in MON compared to retina and brain tissue samples (p = 0.0013 and 0.0060, respectively; Fig 6C).

## Amyloid precursor protein axonal transport

APP antibody labeling was used to assess axonal transport obstruction in B6 and AQP4 null 3D-IOP eyes, compared to contralateral control eyes, using mean fluorescent intensity, mean

**Table 3. AQP4 Pixel Intensity Values (PIV) were quantitatively measured across 7 regions in C57BL/6 control samples.**

| C57BL/6 Controls | Choroid | Retina | PL | Anterior UON | Posterior UON | MTZ |
|---|---|---|---|---|---|---|
| Retina | **p ≤ 0.001** | | | | | |
| PL | p = 0.67 | **p ≤ 0.001** | | | | |
| Anterior UON | p = 0.90 | **p ≤ 0.001** | p = 0.61 | | | |
| Posterior UON | **p ≤ 0.05** | **p ≤ 0.001** | **p ≤ 0.05** | **p ≤ 0.05** | | |
| MTZ | **p ≤ 0.001** | **p ≤ 0.001** | **p ≤ 0.01** | **p ≤ 0.001** | **p ≤ 0.01** | |
| MON | **p ≤ 0.001** | **p ≤ 0.01** | **p ≤ 0.01** | **p ≤ 0.01** | **p ≤ 0.01** | p = 0.14 |

PL = pre-lamina, anterior-UON = anterior-unmyelinated optic nerve, posterior- UON = posterior-unmyelinated optic nerve, MTZ = myelin transition zone,
MON = myelinated optic nerve, Regional group comparisons were made using non-parametric statistical analysis (p-values).

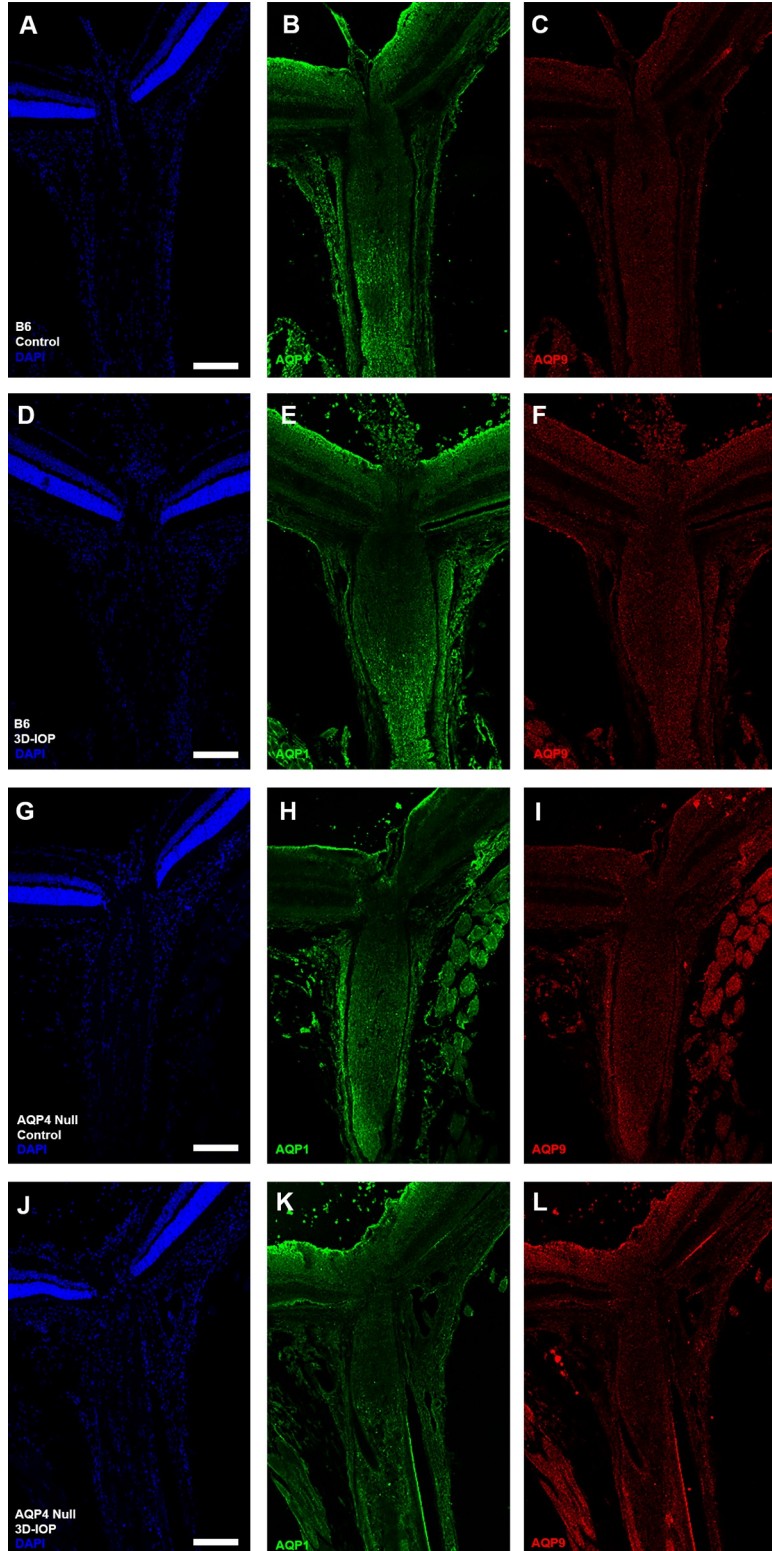

**Fig 4. Ten C57BL/6 (B6) and nine aquaporin 4 (AQP4 null) eyes labeled with antibodies for AQP1 (green), AQP9 (red) and DAPI (blue) in both control B6 (A—C) and AQP4 null (D—F), as well as after 3D-IOP in B6 (G—I) and AQP4 null (J—L).** Scale Bar: 150 μm (A-L).

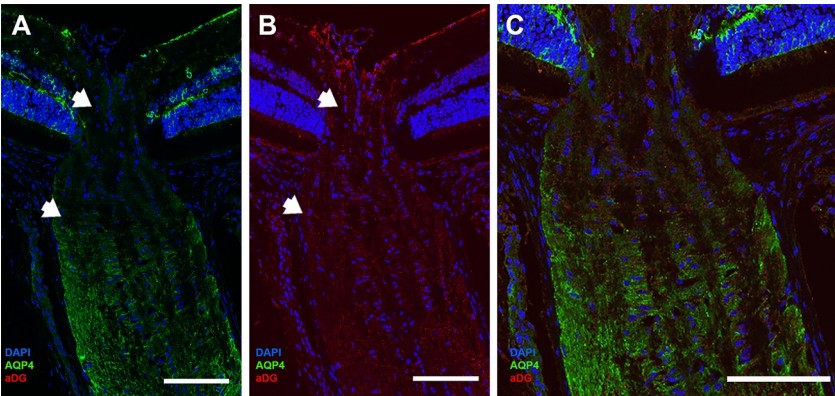

**Fig 5. Normal mouse ONH region labeled for AQP4 (green) and α-dystroglycan (αDG, red, n = 10).** (A) AQP4 is present in the superficial retina, outer plexiform layer of the retina, the MTZ, and MON. AQP4 is minimal present in the PL and UON regions of the ONH (zone between white arrows). (B) αDG is found throughout the retina and all areas of the optic nerve, including the UON where AQP4 is absent. (C) Magnification and merged imaged of AQP4 and αDG staining. Scale Bar: 50 μm (A,B), and 25 μm (C).

intensity of brightest pixels, and fraction of brightest pixels in 4 regions: retina, PL, UON and MON (Fig 1A). In both B6 and AQP4 null 3D-IOP eyes, there were significant increases in the parameters indicating localized transport obstruction (brightest pixel mean intensity and fraction of brightest pixels), but the increases were not significantly different between the two mouse types (S2 Fig).

There were no statistical differences in mean APP fluorescent intensity between strains or regions. Mean intensity of brightest pixels and fraction of brightest pixels were not significantly different between control B6 and control AQP null nerves in retina, PL, or MON.

## Cell proliferation

Both B6 and AQP4 null ON had significant increases in Ki67 labeled nuclei after 3D-IOP in PL, UON, MTZ and MON (Table 4, Fig 7). Bilaterally untreated (naïve) and contralateral controls had no mitotic cells (labeling frequency similar to background level). We have previously

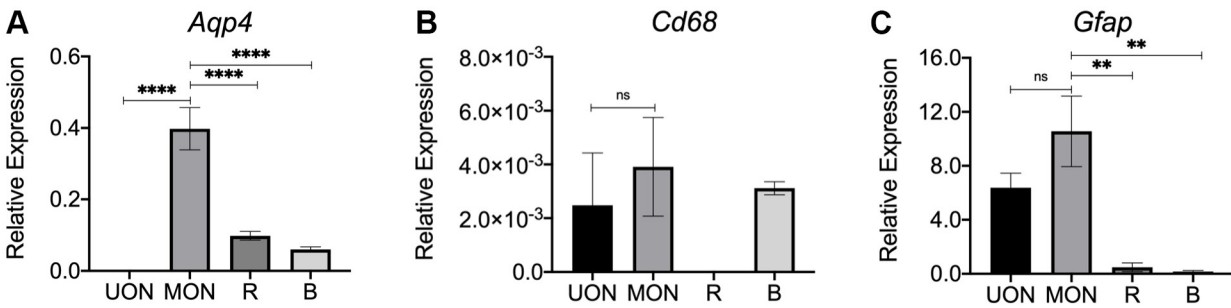

**Fig 6.** A-C) mRNA expression was measured by quantitative real-time PCR in unmyelinated nerve (UON, black bar), myelinated nerve (MON, medium gray bar), retina (R, dark gray bar), and brain (B, light gray bar) from naïve wild-type mice of FVB/N-Tg(GFAP-GFP)14Mes strain to study regional differences in marker genes: (A) *Aqp4*, (B) *Cd68* (microglial marker), and (C) *Gfap* (astrocytes and retinal Müller cells marker). Relative expression was calculated in reference to the geometric mean of housekeeping genes. *Aqp4* levels in the MON were significantly greater compared UON, retina, and brain tissue. No statistically significant differences were found in *Cd68* expression among the different tissues. *Gfap* mRNA levels were largely comparable between UON and MON, but significantly higher in the MON compared to retina or brain regions. Error bars represent SEM; n = 3–6; ns = not significant, ** p < 0.01, ***** p < 0.001, one-way ANOVA with Tukey post-hoc test.

**Table 4. Ki67 positive cells after 3 day IOP elevation.**

| | B6 (cells/mm$^2$) | | AQP4 null (cells/mm$^2$) | | Strain Differences in Controls |
|---|---|---|---|---|---|
| Region | CONTROL | 3 DAY | CONTROL | 3 DAY | By location |
| Pre-Lamina | 0.003 ± 0.01 | 0.44± 0.31*** | 0.01 ± 0.02 | 0.19± 0.12** | **p ≤ 0.03** |
| Unmyelinated ON | 0.01 ± 0.00 | 1.00 ± 0.44*** | 0.01 ± 0.02 | 0.70 ± 0.58*** | p = 0.22 |
| Myelin Transition Zone | 0.003 ± 0.01 | 0.63 ± 0.34*** | 0.01 ± 0.04 | 0.30 ± 0.28** | **p ≤ 0.04** |
| Myelinated ON | 0.01 ± 0.02 | 0.34 ± 0.20 *** | 0.00 ± 0.00 | 0.09 ± 0.08** | **p ≤ 0.01** |
| Number | 9 | 9 | 10 | 10 | |

Data are mean ± standard deviation (STDV).

* p ≤ 0.05

** p ≤ 0.01

***p ≤ 0.001, unpaired t-tests: control vs. 3 day.

reported significant proliferation in UON after 3 day and 1 week IOP elevations, followed by return to control Ki67 labeling after 6W-GL [39]. B6 ON had significantly more Ki67 labeled cells than AQP4 null in PL, MTZ and MON, but Ki67 labeling in UON was not significantly different between the two strains (p ≤ 0.03, p ≤ 0.04, p ≤ 0.01, and p = 0.22, respectively, unpaired t-tests, Table 4).

We performed double labeling with antibodies to Ki67 and either GFAP (astrocytes) or Iba1 (microglia). After 3D-IOP, there were both astrocytes and microglia among the proliferating cells, and Ki67 positive cells were most frequent in the UON region (Figs 7 and 8). Co-labeling of proliferating astrocyte cells were more common than co-labeled microglia, though

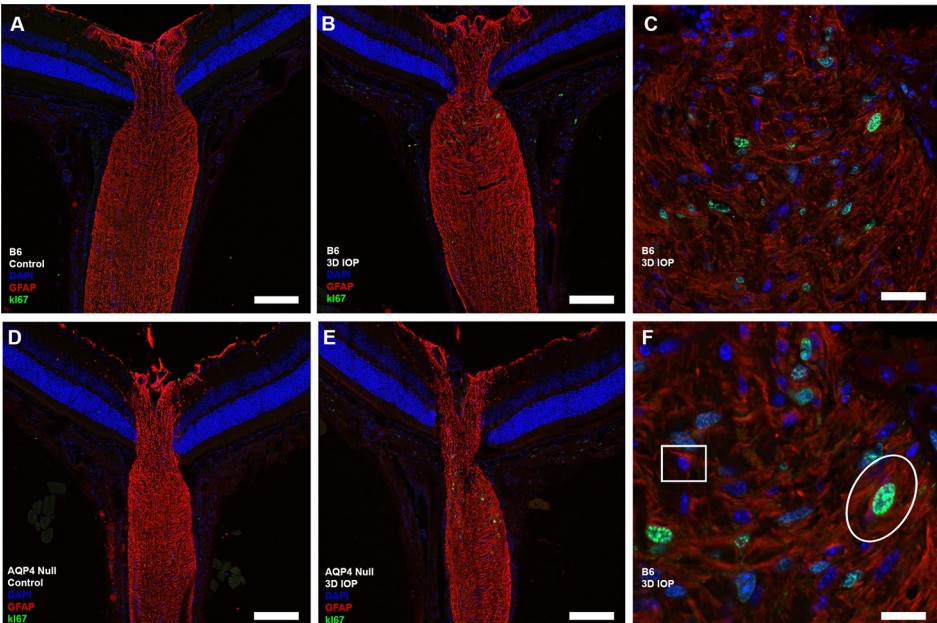

**Fig 7. Cryopreserved longitudinal sections were labeled with antibodies to glial fibrillary acidic protein (GFAP, red), cell proliferation marker (Ki67, green) and DAPI (blue).** Ten C57BL/6 (B6, A) and nine aquaporin 4 null (AQP4 null, D) controls showed no positive Ki67 cells. Significant cell proliferation was seen after 3D-IOP in B6 (B,C, F) and AQP4 null (E). (F) Normal, non-proliferating astrocyte is outlined by white square, its nucleus surrounded by cytoplasmic GFAP label. A proliferating astrocyte is outlined by white oval, its green nucleus surrounded by cytoplasmic label for GFAP. Scale Bar: 100 μm (A,B,D,E), 40 μm (C), and 10 μm (F).

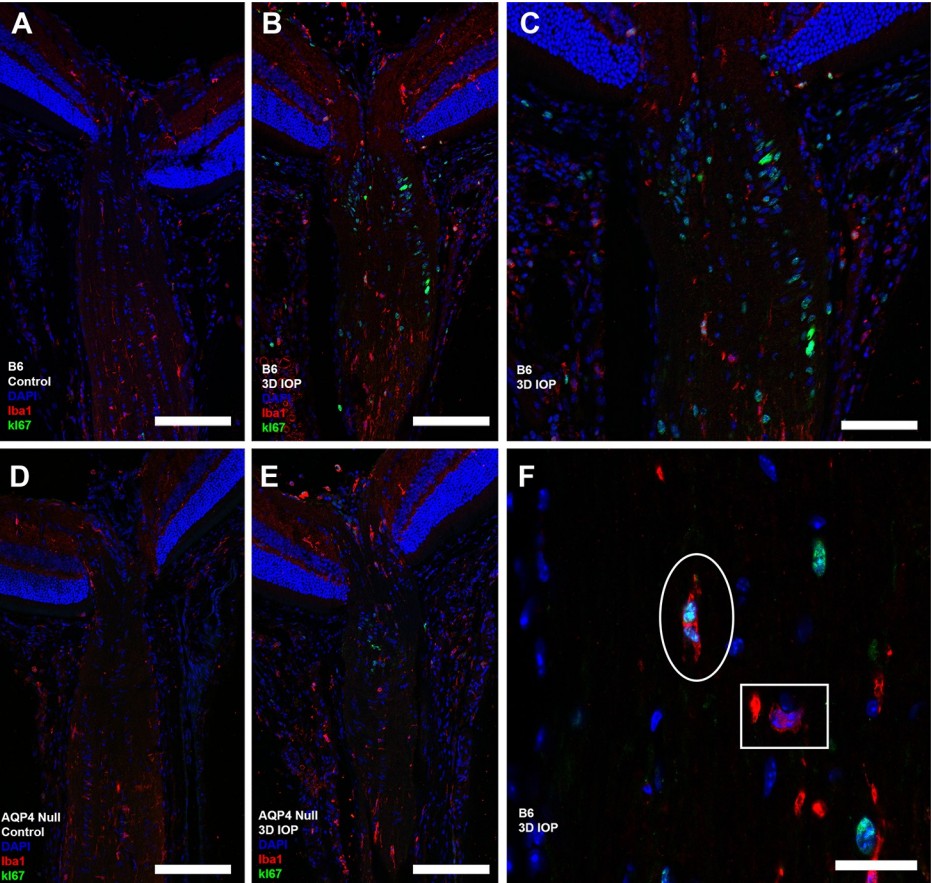

**Fig 8. Cryopreserved longitudinal sections labeled with antibodies to Iba1 (microglia, red) and Ki67 (green), and DAPI label (blue).** Ten C57BL/6 (B6, A) and nine aquaporin 4 null (AQP4 null, D) controls showed no visible Ki67 labeling. Microglia proliferation was visible after 3D-IOP in ten B6 (B,C,F) and nine AQP4 null (E). (F) Normal microglia (white square, nucleus surrounded by Iba1 labeled cytoplasm) and proliferating microglia (white oval, Ki67 positive nucleus surrounded by Iba1 label). Proliferating microglia were more frequently observed in the myelinated optic nerve. Scale Bar: 100 μm (A,B,D,E), 50 μm (C), and 10 μm (F).

the number of each was low enough that statistical analysis was not feasible. Proliferating astrocytes were more often seen in the UON region, while proliferating microglia were more frequent in MON region.

### Retinal ganglion cell axon loss after 6 week IOP elevation

Assessment of mean ON area, axon density, axon number and axon diameter for B6 and AQP4 null animals in untreated controls and after 6W-GL are shown in Table 5. Mean ON area in controls was 27% smaller in AQP4 nulls (p ≤ 0.01) than B6, but the number of axons was not significantly lower in AQP4 nulls (Table 5). Mean total number of axons was similar in control eyes of the two strains, 47,140 axons in B6 and 43,586 axons in AQP4 nulls, but mean axon density was significantly higher in AQP4 null controls than in B6 controls (660,811 vs. 523,951 axons/mm$^2$, p ≤ 0.001, unpaired t-test). Mean nerve axon diameter was 4.7% smaller in AQP4 null than B6 (p = 0.01, unpaired t-test).

After 6W-GL, mean axon loss compared to their controls was 23.3% in B6 and 24.6% in AQP4 null mice (both p ≤ 0.001, unpaired t-test). There was no statistical difference between axon loss in the two strains (p = 0.80; unpaired t-test, Table 5). In multivariable analysis, the

**Table 5. Optic nerve area and axon number after 6 week IOP elevation.**

|  | ON Area Control (mm$^2$) | ON Area Glaucoma (mm$^2$) | ON Area Loss (%) | Density Loss (%) | Axon Number Control | Axon Number Glaucoma | Mean Axon Diameter Control (μm) | Mean Axon Diameter Glaucoma (μm) |
|---|---|---|---|---|---|---|---|---|
| C57BL/6 | 0.092 ± 0.02 | 0.084 ± 0.01 | 8.1 | 16.4 | 47,140.1 ± 10274.3 | 36,433.4 ± 11,593.8 | 0.322 ± 0.03 | 0.322 ± 0.03 |
| AQP4 Null | 0.067 ± 0.01 | 0.061 ± 0.01 | 5.1 | 15.4 | 43,586.8 ± 10451.9 | 32,856.8 ± 9,488.7 | 0.307 ± 0.03 | 0.311 ± 0.03 |
| B6 vs AQP4 Null | **p ≤ 0.01** | **p ≤ 0.04** | p = 0.55 | p = 0.89 | p = 0.11 | p = 0.18 | **p ≤ 0.01** | p = 0.17 |

B6 = C57BL/6, AQP4 null = Aquaporin 4 null, ON = optic nerve. Mean ± standard deviation, p-values = unpaired t-tests.

difference in axon loss between B6 and AQP4 mice remained insignificant when the individual eye IOP exposure was included in the model (axon loss difference by mouse type, p = 0.60). Likewise, multivariable analysis comparing axon loss by mouse strain, IOP exposure, and sex found no significant difference in axon loss based associated with these variables (p = 0.73).

ON area in both strains at 6W-GL was significantly reduced from their control value (p ≤ 0.001 and p ≤ 0.001, unpaired t-test, respectively), but remained 26% smaller in AQP null than B6 mice (Table 5). At 6W-GL, mean axon density decreased by 16.4% in B6 and by 15.4% in AQP null, mean axon diameter was not significantly different from control in either strain, and mean axon diameter was not different between the strains (Table 5).

We calculated the area and density occupied by axons, excluding myelin sheath, and divided these values by overall ON area in MON for each condition. Axons occupied 4.2 ± 0.8% of ON area in B6 control and 3.5% of area in B6 6W-GL, a 15% reduction (p ≤ 0.001, unpaired t-test). In AQP4 null, axons accounted for 5.3% ± 1.1% of ON area in controls and 4.4% ± 1.1% in 6W-GL samples, an 18% reduction (p ≤ 0.002, unpaired t-test). The proportion of ON area occupied by axons was greater in both AQP4 null control and 6W-GL samples than in their B6 counterparts (p ≤ 0.001 and p ≤ 0.001, respectively, unpaired t-test). We measured the myelin sheath thickness in TEM images, finding no significant difference between control B6 and AQP4 null: 134.5 ± 32.6 nm vs. 136.3 ± 33.5 nm (p = 0.30, unpaired t-test).

## Axial measurements

At baseline, AQP4 null eyes were not significantly longer than B6 (3.4 ± 0.2 mm vs 3.4 ± 0.2 mm, p = 0.33, unpaired t-test), but were somewhat longer after 6W-GL (3.7 ± 0.3 mm vs. 3.5 ± 0.3 mm, p ≤ 0.001, unpaired t-test). B6 mouse eyes exposed to 3D-IOP increased in length by 8.8% and 9.7% in width (both p ≤ 0.001, unpaired t-tests) and in 6W-GL eyes, B6 length increased by 2.8% (p ≤ 0.05, unpaired t-tests) and width increased by 5.7% (p ≤ 0.001, unpaired t-tests). AQP4 null eyes exposed to 3D-IOP increased in length by 6.8% (p ≤ 0.001, unpaired t-tests) and in width by 7.5% (p ≤ 0.001, unpaired t-tests), while in 6W-GL the length increased by 12.3% (p ≤ 0.001, unpaired t-test) and width by 10.3% (p ≤ 0.001, unpaired t-test).

## Astrocyte ultrastructure

Observations of the ultrastructural appearance of astrocytes of the UON were made in control B6 and AQP4 null animals. The astrocyte processes at the junctional area with the peripapillary sclera were not different in appearance between the two strains of mice. Features identified in a recent publication (Quillen et al. 2020 [39]) were similar, including multiple infoldings of

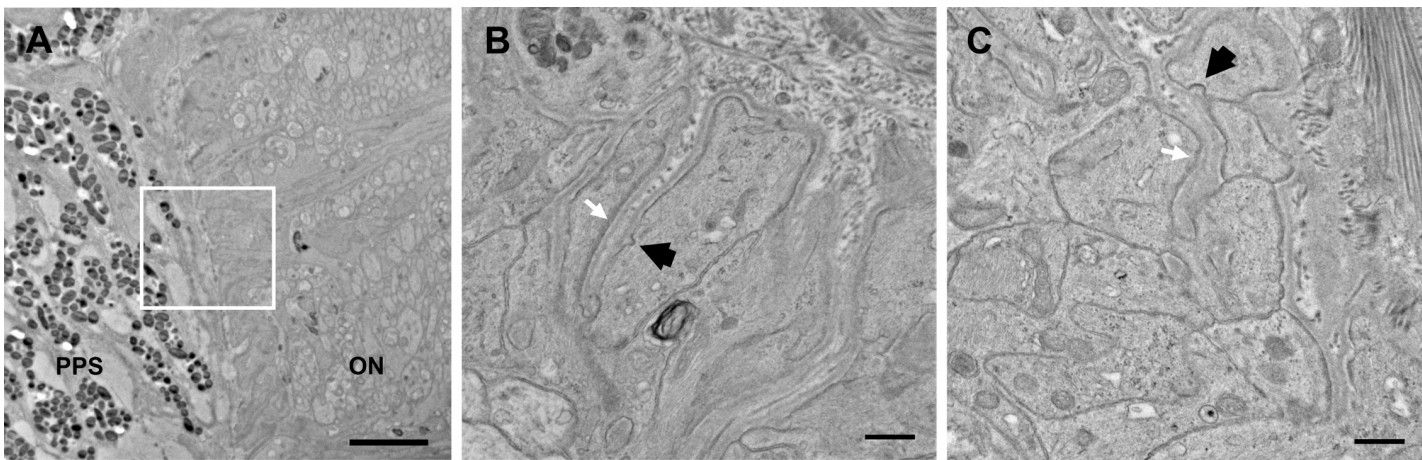

**Fig 9.** (A) Astrocyte junctions with the peripapillary sclera (PPS) in the unmyelinated zone (UON) can be seen by transmission electron microscopy. Higher magnification images (taken at locations indicated in A by a white square) show multiple invaginations of astrocyte processes that are present in both bilaterally naïve control C57BL/6 eyes (B6, B) and bilaterally naïve aquaporin 4 null eyes (AQP4 null, B). A continuous basement membrane of the astrocytes separates it from the sclera and at this site, astrocytes exhibit an electron dense junctional complex (white arrows). Pinocytotic vesicles are seen (black arrows). These features are similar in both mouse strains. Scale bar A = 2um, Scale bar B,C = 500 nm.

astrocyte processes as they join their basement membrane in the zone adjoining the choroid and sclera (Fig 9). Dense junctional complexes were seen along the astrocyte border facing their basement membrane, but not elsewhere in the membrane of astrocyte processes. There were occasional pinocytotic vesicles in astrocyte processes at the scleral junction zone. Gap junctions between astrocytes were often present in both mouse strains.

## Discussion

AQP channels facilitate water movement in the central nervous system, with AQP4 found most prominently. Both AQP1 and AQP4 have channel dimensions sufficient only for passage of water molecules, while AQP9 permits movement of glycerol and urea [1]. Removal of both AQP4 alleles leads to a variety of alterations, including a decrease in the voltage of the electro-retinogram without detectable alteration of retinal structure nor alteration of the blood—brain barrier [25]. AQP4 null mice have reduced inflammatory responses in experimental allergic encephalomyelitis [40], reduced astrocytic migration and scar formation [41], enhanced dextran diffusion in the extracellular space [42], and reduced cerebral edema after brain injury [43, 44]. Recent investigations have suggested that AQP4 deletion could impair the removal of fluid and metabolites and that AQP functions could be involved in glaucoma pathogenesis [34, 45]. Prior to the present report, the effect of AQP deletion had not been evaluated for its effect on loss of retinal ganglion cells in experimental glaucoma models.

In a well-established AQP4 null mouse strain, we demonstrated that the absence of AQP4 channels did not alter either the axonal transport obstruction from 3 day IOP elevation or the degree of ganglion cell axon loss experienced after 6 week IOP increase. With similar IOP elevation levels, AQP4 null mice did not have either a beneficial or a detrimental effect on experimental glaucoma parameters. Furthermore, the degree of glial proliferation in response to IOP elevation was unchanged. Nor did we see evidence that AQP channels other than AQP4 substituted structurally for the absence of this gene in the mice. While it is clear that astrocytic reaction in glaucoma involves a variety of changes in these glia, the participation of AQP4 channels in fluid movements in the retina and optic nerve seems not to play a vital role in experimental glaucoma damage.

In this regard, it is intriguing that the AQP4 channel is minimally present in the astrocytic lamina (UON) of the mouse and rat eye. Nor are the other aquaporins (AQP1 and AQP9) found at the UON region, which is the site of injury to retinal ganglion cell axons in glaucoma. Furthermore, there is no change in the presence of AQP4 in the UON with IOP elevation in mouse. It is known that the formation of orthogonal arrays of AQP particles at the cell membrane into channels is dependent on the presence of the dystroglycan complex of transmembrane molecules, including αDG. We found that αDG was present in the astrocytic membranes throughout the retina and ON, including the UON. Thus, AQP4 is not minimized at this location due to lack of its supportive dystroglycan complex. This evolutionarily conserved regional lack of AQP4 at the UON has potentially important implications for how the ON is protected in glaucoma. The simplest conclusion is that the minimal presence of AQP4 at the PL and the UON is somehow advantageous in protecting ganglion cell axons from damage. There are several possible explanations for such a beneficial phenotype.

First, the lack of AQP4 at the UON could reduce local tissue edema by preventing imbibition of water into astrocytes. By failing to expand in the closed ONH compartment, astrocytes could avoid enhancing the axonal transport obstruction in axons. In the brain, retina, and myelinated ON, astrocytes contact capillaries and other vessels or the brain surface at all of their processes, but do not form a basement membrane to contact connective tissues. By contrast, UON astrocytes (and those in the lamina cribrosa of larger mammals) are widely in contact with the dense connective tissue of the peripapillary sclera (or of lamina cribrosa beams). As a result, their topography is different from astrocytes in other areas that reside entirely within the neuropil. Speculatively, UON astrocytes might find imbibed water difficult to discharge at their peripheral basement membrane zone. Thus, the absence of AQP4 channels would avoid detrimental water intake.

A second potential explanation for the lack of AQP4 at the UON relates to the fact that UON astrocytes in mouse and in larger mammals are uniquely subjected to asymmetric mechanical forces from hoop stress generated in the peripapillary sclera and the translaminar pressure gradient from inside the eye outward. No other astrocyte has a similar need to mechanically sense stress and to respond to it in a manner that preserves normal neuronal structure and function. Hypothetically, the lack of water ingress from scarce presence of AQP channels could permit more robust cytoskeletal support of the tissue that might be compromised by intracellular volume expansion.

A final possible reason for relative absence of AQP4 at the UON is the known absence of the blood—brain barrier at the choroid, which permits cytokines and other proteins access to the ONH [46]. Without AQP4 channels to imbibe fluid, extracellular diffusion out of the UON might speed the removal of such chemical and protein elements from the ONH and be protective of RGC axons. Whatever the reasons for the minimal AQP4 presence at the ONH, it is one of a growing set of unique phenotypical features of the astrocytes of this local zone.

There are a number of reports that attempt to quantify changes in gene expression of AQP in mouse and rat eyes with IOP elevation. While no gene changes were noted after one hour IOP increase in mouse [47], optic nerve crush was reported to lead to decreased AQP4 expression, while in DBA/2J mouse glaucoma, AQP4 rose in a small number of nerves [25]. Johnson et al [30] reported that AQP4 expression decreased by qPCR in 5 week rat glaucoma. Others have reported AQP4 upregulation after ON crush [8, 9]. Careful examination of the precise tissues that were included in the above studies shows that none segregated the UON from the PL or MTZ/MON portions of the nerve, and each included either PL astrocytes, MTZ/MON astrocytes, or both. Examples of the variety of tissues from which astrocytes were studied by others include: unmyelinated ONH and MTZ [25], prelaminar ONH, unmyelinated ONH, and MTZ [30], unmyelinated ONH segment including the pia and pigmented peripapillary

border tissue [48], central retina as well as a small portion of choroid, sclera and MTZ [49] and retinal astrocytes and Müller cells [50]. Hence, all past studies, to our knowledge, have not determined whether changes in AQP4 occurred in portions of the ON that normally express it or in the key area that normally does not. We found no change in AQP4 labeling in the UON region with IOP elevation.

AQP4 null mice had similar axial length and width, as well as similar baseline IOP compared to B6 mice. They also responded to microbead injection with similar IOP elevations and axial globe enlargement. Likewise, the two strains showed similar cell proliferation of both astrocytes and microglia. Intriguingly, control AQP4 null MON studied by light and electron microscopy found some notable differences from B6. Their ON areas were substantially smaller, with higher axon density, but similar number of axons, mean axon diameter and mean myelin thickness. Any differences in axon number and diameter could account for only a tiny proportion of the smaller ON area and greater density of axons, so the lower ON area in AQP4 is most likely due to lower astrocyte area, perhaps related to the absence of AQP4 channels. By analogy, it suggests that the astrocyte process area of a zone without AQP4 channels would have lower astrocyte area. Indeed, in quantitative analysis of astrocyte area as a proportion of ON tissue in fixed, cryosectioned mouse tissue, we found that the astrocyte area in the UON, which is devoid of AQP4, is less than half that of the MON [51].

Astrocytes are the most numerous glia cell in mammalian ONH [52] and the reaction of astrocytes to injurious stimuli is now recognized to be either beneficial or detrimental [53–55]. Designated as A2 astrocytes, the beneficial phenotype activates the JAK-STAT pathway [56] and STAT knockout in mouse glaucoma leads to greater ganglion cell loss [25]. The detrimental A1 astrocyte phenotype arises from stimulation from microglia with interleukin 1α, tumor necrosis factor α (TNFα), and complement component subunit 1q (C1q). Inhibition or knockout of TNFα [57] or C1q [58] reduce experimental glaucoma loss of ganglion cells. This suggests that both types of astrocytes may be involved in the response to glaucoma and that the relative equilibrium between them may help to determine the degree of damage. In this study, we confirmed the proliferation of both microglia and astrocytes in the UON of mouse after 3 days of IOP elevation. While some investigators have stated that most proliferating cells in the glaucoma ON are Iba1 positive microglia [25], our findings agree with those of Lozano and co-workers that, while microglia and astrocytes both proliferate [56], astrocytes remain the majority of glia in the glaucomatous ON. In chronic experimental glaucoma in monkey, the total number of glia appeared constant, though the proportionate area of glia increased with axonal loss [59]. The interplay between microglia and astrocytes in the UON and lamina cribrosa is an important area for continued study.

The presence of a glymphatic pathway for fluid exit from the brain was proposed by Iliff et al. [60], based on movements of various solutes. While not disagreeing that there is extracellular movement of solutes, Verkman and co-workers have challenged the interpretation of these data and the existence of an astrocytic-dependent pathway [61], finding no difference in amyloid β movement between AQP4$^{+/+}$ and AQP null mice, the same strain used here. The controversy over whether astrocytes are involved in a glymphatic pathway includes advocates who have provided confirming data [62], as well as alternative interpretations continuing to deny its existence [63]. Tracer studies by Matthieu et al. [35] showed appearance of dyes in the MON after cisternum magnum injection, but did not document that astrocytes were a direct part of the movement [28]. Since only water molecules pass through AQP4 channels, the movement of these tracers could not be through them. After RGC axonal loss in DBA/2J mouse glaucoma after months of exposure, the movement of tracers in the myelinated optic nerve was altered, both in Wang et al. and Matthieu et al [34, 45]. However, the cellular and extracellular content of atrophic optic nerves is substantially different from normal, leaving

several explanations for changes in solute movement. It has been proposed that abnormal glymphatic pathway behavior could be a contributing feature of glaucoma damage [64]. If such a pathway exists and if it depends upon AQP4 channels as proposed, our data do not support this hypothesis.

In summary, we found that mice without AQP4 have smaller ON area, but similar numbers of retinal ganglion cell axons than B6 mice, in all likelihood due to smaller proportion of the ON occupied by astrocytes in the AQP4 null mice. We confirmed previous publications that AQP4 is minimally present in the UON of mice, despite the presence of α-DG in the astrocyte membranes. Lack of AQP4 conferred no beneficial or detrimental effect on either short-term axonal transport blockade or ganglion cell loss in experimental IOP elevation. We speculate that its universally minimal presence at this location is a potential evolutionary safeguard against axonal injury.

## Supporting information

**S1 Fig. Schematic of mouse optic nerve head regions denoted and anti-aquaporin-4 pixel intensity brightness comparison between each.** Schematic shows cryopreserved C57BL/6 control (B6) optic nerve head tissue stained with DAPI (blue) with outlined regions chosen for pixel intensity brightness analysis. First schematic (A) was divided into outer (orange) and inner (green) regions at the pre-lamina (PL, from vitreoretinal surface to a line joining the two endpoints of BMO), anterior unmyelinated optic nerve (anterior UON, from BMO to 100 μm posteriorly), posterior unmyelinated optic nerve (posterior UON, from 100 μm to 200 μm posteriorly), myelin transition zone (MTZ, from 200 μm to 350 μm posteriorly), and the myelinated optic nerve (MON, from 350 μm to the end of the section). Background area (white circle, A) and choroid (white rectangle, B) were used as AQP4 negative controls. Second schematic (B) shows the total area (gray outlines) of the regions; retina, choroid, PL, anterior-UON, posterior-UON, MTZ and MON. (C) Mean AQP4 pixel intensity value (PIV) graph plots 8 regions calculated using FIJI software in B6 control nerves immunostained for AQP4 in the three areas; outer area (orange bars), inner area (green bars) and total area (gray bars) as defined in (A) and (B). Standard error bars are plotted. Gray dotted line identifies the AQP4 background level in choroid. Scale Bar: 100 μm (A,B).
(TIF)

**S2 Fig. Pixel intensity value of mouse optic nerve head regions stained for amyloid precursor protein.** Amyloid precursor protein (APP) labeling assessed axonal transport block at 3D-IOP in four regions of the mouse optic nerve head: retina, pre-lamina (PL), unmyelinated optic nerve (UON) and myelinated optic nerve (MON). B6 (red) and AQP4 null (blue) nerves had similar transport block at PL and UON in the two key metrics, mean intensity of brightest pixels at the 97.5[th] percentile (A) and fraction of brightest pixels at the 97.5% percentile (B).
(TIF)

## Acknowledgments

We would like to thank Peter Agre, Ole Petter Ottersen, and Michael Levy for providing AQP4 null mice and for their advice and support. We also thank Joan Jeffrey for her statistical analysis, and both Don Zack and Thomas Johnson for advice and manuscript review.

## Author Contributions

**Conceptualization:** Elizabeth Kimball, Mary Ellen Pease, Harry Quigley.

**Data curation:** Elizabeth Kimball, Julie Schaub, Sarah Quillen, Casey Keuthan, Mary Ellen Pease, Harry Quigley.

**Formal analysis:** Elizabeth Kimball, Julie Schaub, Sarah Quillen, Casey Keuthan, Mary Ellen Pease, Arina Korneva, Harry Quigley.

**Funding acquisition:** Elizabeth Kimball, Harry Quigley.

**Investigation:** Elizabeth Kimball, Julie Schaub, Mary Ellen Pease, Arina Korneva, Harry Quigley.

**Methodology:** Elizabeth Kimball, Julie Schaub, Sarah Quillen, Casey Keuthan, Mary Ellen Pease, Harry Quigley.

**Project administration:** Elizabeth Kimball, Harry Quigley.

**Resources:** Elizabeth Kimball, Harry Quigley.

**Software:** Elizabeth Kimball, Harry Quigley.

**Supervision:** Elizabeth Kimball, Harry Quigley.

**Validation:** Elizabeth Kimball, Julie Schaub, Harry Quigley.

**Visualization:** Elizabeth Kimball, Harry Quigley.

**Writing – original draft:** Elizabeth Kimball, Julie Schaub, Sarah Quillen, Casey Keuthan, Mary Ellen Pease, Arina Korneva, Harry Quigley.

**Writing – review & editing:** Elizabeth Kimball, Julie Schaub, Sarah Quillen, Casey Keuthan, Mary Ellen Pease, Arina Korneva, Harry Quigley.

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
