## [Decision Letter · Decision Letter 0]

29 Dec 2020

PONE-D-20-37793

The Role of Aquaporin-4 in Optic Nerve Head Astrocytes in Experimental Glaucoma

PLOS ONE

Dear Dr. Kimball,

Thank you for submitting your manuscript to PLOS ONE. After careful consideration, we feel that it has merit but does not fully meet PLOS ONE’s publication criteria as it currently stands. Therefore, we invite you to submit a revised version of the manuscript that addresses the points raised during the review process.

The reviewers have recommended minor revision and have offered constructive recommendations. 

We look forward to receiving your revised manuscript.

Kind regards,

Sanjoy Bhattacharya

Academic Editor

PLOS ONE

Journal Requirements:

Reviewers' comments:

Reviewer's Responses to Questions

**Comments to the Author**

1. Is the manuscript technically sound, and do the data support the conclusions?

Reviewer #1: Yes

Reviewer #2: Yes

2. Has the statistical analysis been performed appropriately and rigorously? 

Reviewer #1: Yes

Reviewer #2: Yes

3. Have the authors made all data underlying the findings in their manuscript fully available?

Reviewer #1: Yes

Reviewer #2: Yes

4. Is the manuscript presented in an intelligible fashion and written in standard English?

Reviewer #1: Yes

Reviewer #2: No

5. Review Comments to the Author

Reviewer #1: The manuscript demonstrates that aquaporin does not have beneficial nor detrimental effects in a mouse model of glaucoma. This information provides insight into the distribution of aquaporin in the retina, optic nerve disk and optic nerve. The authors use an animal mouse model of glaucoma by inducing elevated intraocular pressure via bead occlusion. This glaucoma model was implemented in C57BL6 controls and AQP4 null mice. The authors used fluorescent immunohistochemistry to analyze the distribution of aquaporins in the eye and optic nerve and quantitative real-time PCR for measuring mRNA expression of aquaporins. Axonal transport was assessed through the movement of amyloid precursor protein.

Consider revising this sentence in introduction. Three AQP types have been described in astrocytes; AQP1, AQP4 and AQP9, aquaglyceroporin.

Include a brief overview of how the bead injections were carried out including vehicle used and where the microbeads were procured from (company).

Specify what type of knock-out was used (constitutive, conditional)?

Reviewer #2: Although this paper contains some important information on aquaporin channel expression in a mouse model of glaucoma, it requires minor revision. The manuscript should be revised with proper English grammar.

6. PLOS authors have the option to publish the peer review history of their article (what does this mean?). If published, this will include your full peer review and any attached files.

Reviewer #1: No

Reviewer #2: No

---

## [Author Response · Author response to Decision Letter 0]

6 Jan 2021

Manuscript meets PLOS ONE's style requirements.

Reviewer #1: The manuscript demonstrates that aquaporin does not have beneficial nor detrimental effects in a mouse model of glaucoma. This information provides insight into the distribution of aquaporin in the retina, optic nerve disk and optic nerve. The authors use an animal mouse model of glaucoma by inducing elevated intraocular pressure via bead occlusion. This glaucoma model was implemented in C57BL6 controls and AQP4 null mice. The authors used fluorescent immunohistochemistry to analyze the distribution of aquaporins in the eye and optic nerve and quantitative real-time PCR for measuring mRNA expression of aquaporins. Axonal transport was assessed through the movement of amyloid precursor protein.

Consider revising this sentence in introduction. Three AQP types have been described in astrocytes; AQP1, AQP4 and AQP9, aquaglyceroporin.

- This sentences has been corrected and now reads: “Three AQP types are known to express in astrocytes ; AQP1, AQP4, and AQP9 (also referred to as, aquaglyceroporin which is permeable to glycerol and other small uncharged solutes).

Include a brief overview of how the bead injections were carried out including vehicle used and where the microbeads were procured from (company).

- The paragraph now reads: “One anterior chamber was injected with Polybead Microspheres (Polysciences, Inc., Warrington, PA, USA), suspended in sterile PBS (phosphate buffer saline solution), consisting of 2 µL of 6 µm diameter beads, then 2 µL of 1 µm diameter beads, followed by 1 µL of viscoelastic compound (10 mg/ml sodium hyaluronate, Healon; Advanced Medical Optics Inc., Santa Ana, CA). Injections were made with a 50 µm tip diameter glass cannula, connected to a Hamilton syringe (Hamilton, Inc., Reno, NV). The glass cannula was kept in place for 2 minutes to prevent the egress of beads after withdrawal. The contralateral eye was used as control. No animal was excluded from our study once the bead injection was completed.”

Specify what type of knock-out was used (constitutive, conditional)?

- The AQP4 null animals studied here are now referred to as “Aquaporin 4 constitutive knockout (AQP4 null)”.

Animal numbers have been corrected for the gene expression portion of this manuscript (previously it stated 6 animals were studied, however, the correct total is 10). Both the total animal count, and Table 1 (Mouse Groups) have been amended. 

Reviewer #2: Although this paper contains some important information on aquaporin channel expression in a mouse model of glaucoma, it requires minor revision. The manuscript should be revised with proper English grammar.

- The manuscript has been revised for grammatical errors. 

Thank you,

Elizabeth Kimball

---

## [Editor Report · Decision Letter 1]

12 Jan 2021

The Role of Aquaporin-4 in Optic Nerve Head Astrocytes in Experimental Glaucoma

PONE-D-20-37793R1

Dear Ms. Kimball,

We’re pleased to inform you that your manuscript has been judged scientifically suitable for publication and will be formally accepted for publication once it meets all outstanding technical requirements.

Kind regards,

Sanjoy Bhattacharya

Academic Editor

PLOS ONE
---

## [Editor Report · Acceptance letter]

22 Jan 2021

PONE-D-20-37793R1 

The Role of Aquaporin-4 in Optic Nerve Head Astrocytes in Experimental Glaucoma 

Dear Dr. Kimball:

I'm pleased to inform you that your manuscript has been deemed suitable for publication in PLOS ONE. Congratulations! Your manuscript is now with our production department. 

Kind regards, 

on behalf of

Dr. Sanjoy Bhattacharya 

Academic Editor

PLOS ONE